# Update on Genetics of Primary Aldosteronism

**DOI:** 10.3390/biomedicines9040409

**Published:** 2021-04-10

**Authors:** Kiyotaka Itcho, Kenji Oki, Haruya Ohno, Masayasu Yoneda

**Affiliations:** Department of Molecular and Internal Medicine, Graduate School of Biomedical and Health Sciences, Hiroshima University, Hiroshima 734-8551, Japan; itcho@hiroshima-u.ac.jp (K.I.); haruya-ohno@hiroshima-u.ac.jp (H.O.); masayone17@hiroshima-u.ac.jp (M.Y.)

**Keywords:** primary aldosteronism, hypertension, somatic mutation, aldosterone-producing adenoma

## Abstract

Primary aldosteronism (PA) is the most common form of secondary hypertension, with a prevalence of 5–10% among patients with hypertension. PA is mainly classified into two subtypes: aldosterone-producing adenoma (APA) and bilateral idiopathic hyperaldosteronism. Recent developments in genetic analysis have facilitated the discovery of mutations in *KCNJ5*, *ATP1A1*, *ATP2B3*, *CACNA1D*, *CACNA1H*, *CLCN2*, and *CTNNB1* in sporadic or familial forms of PA in the last decade. These findings have greatly advanced our understanding of the mechanism of excess aldosterone synthesis, particularly in APA. Most of the causative genes encode ion channels or pumps, and their mutations lead to depolarization of the cell membrane due to impairment of ion transport. Depolarization activates voltage-gated Ca^2+^ channels and intracellular calcium signaling and promotes the transcription of aldosterone synthase, resulting in overproduction of aldosterone. In this article, we review recent findings on the genetic and molecular mechanisms of PA.

## 1. Introduction

Aldosterone is synthesized in the adrenal cortex and plays an essential role in regulating blood pressure by promoting sodium reabsorption in the kidney. Primary aldosteronism (PA), which is a disorder of excess aldosterone secretion, is the most common form of secondary hypertension, with a prevalence of 5–10% among patients with hypertension [1]. The risk of cardiometabolic and renal disease is higher in PA patients than in essential hypertension patients; thus, early diagnosis and appropriate treatment of PA are important for reducing its complications [2,3,4,5]. PA is mainly classified into two subtypes: aldosterone-producing adenoma (APA) and bilateral idiopathic hyperaldosteronism (BHA). Although the etiology of PA has long remained unclear, recent developments in genetic analysis, including next-generation sequencing (NGS), have expanded our understanding of the genetic and molecular mechanisms of PA in the last decade. Exome sequencing discovered somatic mutations in *KCNJ5*, *ATP1A1*, *ATP2B3*, *CACNA1D*, *CACNA1H*, *CLCN2*, and *CTNNB1* in APA [6,7,8,9,10,11,12,13]. Most of the causative genes encode ion channels or pumps, and their mutations lead to depolarization of the cell membrane due to impairment of ion transport. Depolarization activates voltage-gated Ca^2+^ channels and intracellular calcium signaling and promotes the transcription of aldosterone synthase (*CYP11B2*), resulting in overproduction of aldosterone (Figure 1). Furthermore, some key molecules such as VSNL1, CALN1, GSTA1, NPNT, and CLGN have been detected in APA, and their functions in aldosterone production have been elucidated [14,15,16,17,18]. Epigenetic regulation of *CYP11B2* has also been indicated in APA [19,20,21,22].

Familial hyperaldosteronism (FH) has also been reported as a rare cause of PA. There are four forms of FH (FH type 1 to type 4). Although it is rare, the study of FH was preferred as an approach to understand the pathophysiology of PA due to its heritability. The first report of FH was the case of a father and a son presenting the symptoms of PA in 1966, which was corrected by glucocorticoid treatment [23]. Thus, this form of PA is called glucocorticoid-remediable aldosteronism (GRA) or FH type 1. In 1992, linkage analysis revealed that the molecular etiology of GRA was a chimeric gene composed of the promoter of 11β-hydroxylase (*CYP11B1*) fused with the coding region of *CYP11B2*, resulting in aldosterone overproduction regulated by ACTH [24]. The chimeric *CYP11B1/CYP11B2* gene was not identified in APA [25], whereas some causative genes, including *KCNJ5*, *CLCN2*, and *CACNA1H*, have been discovered in the other forms of FH [6,10,11,12].

In this review, we aimed to summarize the molecular mechanisms by which genetic mutations mediate aldosterone production and the clinical and pathological findings related to the gene mutations.

## 2. KCNJ5

In 2011, Choi et al. analyzed 22 cases of APA using whole-exome sequencing and identified two recurrent somatic mutations of *KCNJ5* (G151R and L168R) [6]. *KCNJ5* encodes the G protein-coupled inwardly rectifying K^+^ channel (GIRK4), which belongs to GIRK family members (GIRK1 to GIRK4). GIRK4, which consists of two membrane-spanning domains, one pore-forming region between the two transmembrane domains, and intracellular N and C termini, forms a channel as a homotetramer or heterotetramer with GIRK1. Both substitutions are located near the channel’s ion-selective filter and cause depolarization of the cell membrane due to the loss of ion selectivity of the K^+^ channel and the increased intracellular influx of Na^+^. The authors proposed that activated voltage-gated Ca^2+^ channels resulting from these mutations promote autonomous secretion of aldosterone and cell proliferation. In subsequent studies with adrenocortical carcinoma cell lines, introduction of the *KCNJ5* mutation promoted aldosterone synthesis via depolarization of the cell membrane, allowing sodium and calcium influx into the cell [26,27,28,29]. Mutated *KCNJ5* also increased the expression of *CYP11B2* with its transcription factors nuclear receptor related 1 (*Nurr1*) and activating transcription factor 2 (*ATF2*), and these stimulatory effects were inhibited by Ca^2+^ channel blockers [26,27,30]. Moreover, molecules related to calcium signaling, such as VSNL1 and CALN1, are highly expressed in APA, and they have important roles in aldosterone production [14,15,31]. These results show that increased *CYP11B2* expression is mediated by the Ca^2+^/calmodulin cascade. The relationship between *KCNJ5* mutation and cell proliferation is still controversial, and the difference in *KCNJ5* mutation modulation levels may influence adrenal cell growth [26,32,33]. Several other *KCNJ5* mutations such as E145Q, I157del, and T158A have been reported, although G151R and L168R are the most frequent [8,29,34,35,36,37,38,39,40,41,42,43,44,45].

*KCNJ5* is the most commonly mutated somatic gene in Asians, Europeans, and Americans with APA [38,41,45]. In a report of 474 APA cases from the European Network for the Study of Adrenal Tumors (ENS@T), *KCNJ5* mutation was found in 38% of cases [45]. In White Americans and African Americans, *KCNJ5* mutation was found in 43% and 34% of cases, respectively [37,42]. Conversely, reports from East Asia have shown that nearly 70% of APA patients have a *KCNJ5* mutation, with an ethnic difference [41,43,46,47,48,49,50]. A meta-analysis showed that APA patients with *KCNJ5* mutation have phenotypic features of higher plasma aldosterone levels, young age, female sex, and larger tumor size [51]. Subclinical hypercortisolism is sometimes accompanied by APA; aldosterone and cortisol co-producing adenoma has also been reported in *KCNJ5*-mutated APA [52]. However, a recent prospective study showed that subclinical hypercortisolism was common in APA without *KCNJ5* mutation or with a relatively larger tumor size [53]. Cardiovascular complications in APA patients with *KCNJ5* mutations also have been evaluated in some studies. In *KCNJ5*-mutated APA patients, higher left ventricular mass index (LVMI) and plasma aldosterone levels were reported than in those without *KCNJ5* mutation [54]. Another group reported that the *KCNJ5*-mutated group significantly improved LVMI after surgery [55]. A recent study also showed that APA patients with *KCNJ5* mutations had higher LVMI and inappropriately excessive LVMI (ieLVMI), as well as a greater regression of LVMI and ieLVMI after adrenalectomy, in comparison to those without *KCNJ5* mutations in a propensity-score-matched cohort [56]. These results indicate *KCNJ5* mutation is associated with left ventricular remodeling and diastolic function. *KCNJ5* mutation was also reported to be a predictor of hypertension remission after adrenalectomy for APA [43,57]. On the other hand, subclinical hypercortisolism in patients with APA was indicated to be associated with a lower clinical complete success rate after adrenalectomy [53].

The adrenal cortex comprises three morphologically and functionally distinct layers: zona glomerulosa (ZG), zona fasciculata (ZF), and zona reticularis (ZR). Although the expressions of steroid enzymes are zone-specific, the histological features of APA are heterogeneous [58]. CYP11B2 is specifically expressed in ZG, and 17α-hydroxylase/17,20-lyase (CYP17A1) is expressed in ZF and ZR in the normal adult adrenal gland; however, APA with a *KCNJ5* mutation typically has predominant clear cells (ZF-like cells) [59], and expression of both CYP11B2 and CYP17A1 is found within the same tumor [60,61]. Plasma levels of the hybrid steroids 18-oxocortisol and 18-hydroxycortisol have been reported to be higher in APA patients, particularly in *KCNJ5*-mutated APA [62], which could be explained by its ZF-significant phenotype (Figure 2.) [63]. Thus, steroids have been indicated as clinical biomarkers, and steroid profiling can be utilized for differentiating subtypes of PA [64,65,66,67].

Germline mutation in *KCNJ5* also has been identified in FH. In 2008, Geller et al. reported the case of a father and two daughters with a new form of PA [68]. They showed early-onset PA and marked adrenocortical hyperplasia, which did not respond to medical therapy and led to bilateral adrenalectomy. Choi et al. genetically analyzed this family and discovered germline *KCNJ5* mutation responsible for the disease, which was later classified as FH type 3 [6]. Since then, various phenotypes of FH type 3 depending on genotype have been reported; T158A, I157S, E145Q, and G151R are reported to have severe early-onset PA with bilateral adrenal hyperplasia, requiring bilateral adrenalectomy [6,69,70,71]. On the other hand, G151E and Y152C are associated with mild PA with no adrenal abnormalities on computed tomography (CT) scan and can be controlled by mineralocorticoid receptor antagonist (MRA) [71,72,73]. In vitro study demonstrated that transduction of *KCNJ5* G151E leads to profoundly large Na^+^ conductance compared with other mutations, leading to Na^+^-influx-dependent cell lethality [71,72]. Therefore, it is suggested that these marked alterations of channel function prevent the development of adrenal hyperplasia, resulting in a mild clinical phenotype. However, there was a report of the early-onset PA with de novo *KCNJ5* G151R germline mutation and no adrenal enlargement whose symptoms were successfully controlled by MRA, indicating that diverse clinical phenotype in FH type 3 cannot be defined solely by *KCNJ5* genotype [74]. In addition, two cases of early-onset PA possibly caused by mosaicism for *KCNJ5* mutations were reported [75,76].

## 3. ATP1A1

Beuschlein et al. identified a somatic mutation in *ATP1A1* in 16/308 (5.2%) APAs [7], and Azizan et al. found it in 2 of 10 ZG-like APAs without *KCNJ5* mutation [8]. In contrast to *KCNJ5*-mutated APA, APA with *ATP1A1* mutation is more commonly found in males and has histological features of predominant ZG-like cells [7,8]. *ATP1A1* encodes the alpha 1 subunit of Na^+^/K^+^ ATPase, which transports three Na^+^ ions in exchange for two K^+^ ions. The alpha subunit is composed of 10 transmembrane domains (M1–M10) with intracellular N and C termini. Several somatic mutations such as G99R, L104R, V332G, and EETA963S were identified in the M1, M4, and M9 domains [7,8,35]. Mutations in the M1 and M4 domains, which result in alteration of K^+^ binding and loss of pump activity, lead to depolarization of the cell membrane and autonomous secretion of aldosterone [7]. Mutations in the M9 domain affect a supposed Na^+^-specific site, resulting in loss of pump activity [8]. These mutations were suggested to lead to abnormal H^+^ or Na^+^ leakage current, which is a similar mechanism to that of the *KCNJ5* mutation [8]. However, in vitro study using adrenocortical cells demonstrated that mutations in *ATP1A1* induce depolarization of the cell membrane and intracellular acidification due to H^+^ leak, but not an overt increase in intracellular Ca^2+^ [77]. The specific mechanism of this acidification in autonomous aldosterone production has not been clarified.

The frequency of *ATP1A1* mutation determined through Sanger sequencing performed on whole tumor sample DNA was not as high as that of *KCNJ5* reported previously. However, a recently developed sequencing method using targeted NGS performed on DNA extracted from formalin-fixed paraffin-embedded tissues expressing CYP11B2 in immunohistochemistry (IHC) has enabled the more frequent detection of somatic mutations in APA [37]. The CYP11B2 IHC-guided targeted NGS approach identified 5.0–17% of *ATP1A1* mutations in APA cases [37,42,78,79], whereas the frequency of *ATP1A1* mutations was 2.4–8.2% using conventional methods [7,35,38,41,45]. There are few reports of specific clinical characteristics of APA patients with non-*KCNJ5* mutation; one report showed that APA patients with ATPase mutation tended to have more severe hyperaldosteronism compared to those with wild type, although the sample size was small [80].

## 4. ATP2B3

*ATP2B3* encodes the plasma membrane Ca^2+^ ATPase type 3 (PMCA3), which exports calcium ions from the cytoplasm. Beuschlein et al. reported somatic mutation of *ATP2B3* along with that of *ATP1A1* in APA [7]. PMCA3 is also composed of 10 transmembrane domains (M1–M10) with intracellular N and C termini. Most of the mutations identified in APA are deletion mutations located in the specific region of the M4 domain, which is involved in Ca^2+^ binding and ion gating [7,37,38,41,42,45,78,79,81]. This mutation is presumed to cause a major distortion of the Ca^2+^ binding site, impairing the clearance of cytoplasmic Ca^2+^ ions. Subsequent in vitro studies have demonstrated that *ATP2B3* mutation promotes aldosterone production by two different mechanisms: (1) reduction of Ca^2+^ export due to the loss of pump function increases resting Ca^2+^ activity and (2) influx of Na^+^ caused by gain of cation permeability leads to depolarization and activates voltage-gated Ca^2+^ channels [82]. The frequency of *ATP2B3* mutation is relatively low, accounting for 0.6–10% of APA cases [7,35,37,38,41,42,45,78,79]. *ATP2B3* mutation was also frequently found in APA mainly composed of ZG-like cells [58,70,83]. However, a recent study using a quantitative histological analytical approach with digital imaging software showed that *ATP2B3*-mutated APA tended to have clear cell dominant features [61].

## 5. CACNA1D

Scholl et al. identified five somatic *CACNA1D* mutations (G403R and I770M) among 43 APAs without *KCNJ5* mutation [9]. *CACNA1D* encodes a calcium channel voltage-dependent L-type alpha-1D subunit, which contains four repeated domains (I–IV), each with six transmembrane segments (S1–S6). These altered residues locate in the S6 segments lining the channel pore and induce a shift in voltage-dependent gating to a more negative voltage, leading to an increase in intracellular Ca^2+^ levels [9]. However, subsequent studies have shown that somatic mutations in *CACNA1D* are found throughout the gene in APA [84]. Azizan et al. also reported somatic *CACNA1D* mutations in ZG-like APA at the same time [8]. They also reported that *CACNA1D* mutations were associated with small tumor size, but this association was not found in a recent study using the CYP11B2 IHC-guided targeted NGS approach [79]. The CYP11B2 IHC-guided targeted NGS approach identified a large number of *CACNA1D* mutations (14–42%) [37,42,78,79] compared to conventional methods (0.6–10.3%) [38,41,45]. Moreover, *CACNA1D* mutations are most prevalent (42%), followed by *KCNJ5* mutations, in African American patients with APA [42].

Scholl et al. also reported de novo germline *CACNA1D* mutations (G403D and I770M) in two children featuring early-onset PA with seizures and neurologic abnormalities (PASNA). Although several cases of neurodevelopmental disease with *CACNA1D* de novo germline mutations have been reported, only four cases presenting early-onset PA have been described to date [9,85,86]. Treatment with calcium channel blockers (amlodipine and nifedipine) normalized blood pressure in two of these cases [9,86], and CT scan showed no adrenal abnormality in one case [9].

## 6. CTNNB1

*CTNNB1* encodes β-catenin, and its mutation induces constitutive activation of Wnt/β-catenin signaling. Although Wnt/β-catenin signaling plays a crucial role in normal development and maintenance of the adrenal cortex [87], activated Wnt/β-catenin signaling is also observed in APA [88,89]. In addition to ion channels and ATPases, mutations in *CTNNB1* have been reported in APA with 0–5.1% frequency [13,37,42,78,79,90]. The extracellular matrix gene *NPNT*, which is downstream of the Wnt/β-catenin signaling pathway, is upregulated in ZG-like APA, especially with *CTNNB1* mutation. NPNT over-expression increases aldosterone production in adrenal cells [17]. *CTNNB1* mutation has also been found in other adrenocortical adenomas and adrenocortical carcinomas [91]. A previous study showed that transgenic mice with constitutive β-catenin activation in adrenal tumors develop hyperaldosteronism and malignancy [92]. Taken together, these results suggest that *CTNNB1* mutations stimulate ZG cell proliferation and Wnt/β-catenin activation participates in aldosterone production. APA with *CTNNB1* mutation is more common in females and has variable histological features [13,90]. A higher risk of residual hypertension after adrenalectomy in patients with *CTNNB1*-mutated APA was shown in one report [90]. Clinical and histological features of APA harboring each somatic mutation are summarized in Table 1.

## 7. CLCN2

In 1991, Gordon et al. reported six relatives who presented with APA or BHA unresponsive to glucocorticoids [93]. Several other familial cases were reported by the same group, which was defined as FH type 2 [94]. The cause of FH type 2 had been unknown for a long time; in 2018, Scholl et al. identified *CLCN2* R172Q germline mutation as the cause of FH type 2 by performing exome sequencing on these families [11]. They further analyzed 80 other young-onset PAs without known mutations and reported several *CLCN2* germline mutations with a frequency of 9.9% [11]. At the same time, Fernandes-Rosa et al. also analyzed 12 young-onset PAs and discovered *CLCN2* G24D de novo germline mutation [12]. *CLCN2* encodes the inwardly rectifying chloride channel ClC2, which is expressed in many tissues, including the adrenal glands. These mutations cause depolarization of the plasma membrane by promoting efflux of Cl^–^ ions through gain of function and activation of *CYP11B2* transcription from voltage-gated Ca^2+^ channel activity [11,12]. The morphology of the adrenal glands varied from normal to unilateral nodules on CT scan, but aldosterone production was bilateral in the three cases that underwent adrenal venous sampling [11]. Recently, somatic mutations of CLCN2 were reported in sporadic APA, but the frequency is rare [95,96].

## 8. CACNA1H

In 2015, Scholl et al. performed exome sequencing in 40 hypertensive patients who developed PA before the age of 10 years and identified the *CACNA1H* M1549V germline mutation in five patients, which was classified as FH type 4 [10]. This mutation occurred de novo in two patients and was inherited in the remaining three [10]. *CACNA1H* encodes a voltage-dependent Ca^2+^ channel T-type alpha-1H subunit (Cav3.2), which is the second most highly expressed calcium channel alpha subunit after *CACNA1D* in the human adrenal cortex [9]. This mutation reduces the normal inactivation of Cav3.2 compared with wild type and also activates the channel with less depolarization, causing intracellular Ca^2+^ influx, which is a similar mechanism to the *CACNA1D* mutation [10]. They did not show neurodevelopmental symptoms seen in PASNA and adrenal hyperplasia on CT scan, although one sporadic APA case with multiplex developmental disorder and germline *CACNA1H* mutation was reported [10,97]. The clinical and molecular characteristics of FH are summarized in Table 2. In addition, somatic *CACNA1H* mutations were also reported in sporadic APAs without known mutations using the CYP11B2 IHC-guided sequencing approach [78,98].

## 9. Other Genes Described in Patients with PA

Somatic mutation of *PRKACA*, which causes adrenal Cushing’s syndrome, leads to constitutive activation of protein kinase A (PKA), resulting in excess cortisol production [99]. Somatic mutation of *PRKACA* was reported in a patient with aldosterone and cortisol co-secreting adenoma [100]. Somatic mutation of *GNAS*, which also causes adrenal Cushing’s syndrome due to constitutive activation of the PKA/cAMP pathway, was reported in two patients with aldosterone and cortisol co-secreting adenoma [101]. Somatic mutations in both genes were also reported in the subsequent study using CYP11B2 IHC-guided targeted NGS, but those mutations were detected in CYP11B2-negative adrenal tumors from APA patients [37,42]. The role of somatic mutation in *PRKACA* and *GNAS* in the pathogenesis of PA has not been clarified. Genetic variants of *ARMC5*, *ATP2B4*, *PDE2A*, and *PDE2B* were indicated to be associated with BHA [102,103,104,105,106].

## 10. Conclusions and Perspective

Advances in NGS-based analysis techniques over the past decade have revealed that mutations in ion channels and pumps play a profound role in the pathogenesis of many APAs. The CYP11B2 IHC-guided targeted NGS approach has been reported to detect mutations in up to 96% of APA cases [78]. Although these discoveries have shed considerable light on the mechanisms of aldosterone overproduction, the mechanisms of APA growth and tumorigenesis remain largely unknown. In the future, exploiting every technology and skill would facilitate the elucidation of the pathogenesis of APA without any mutations. Further basic research is required to explain the tumorigenesis and cell growth in APA.

## Figures and Tables

**Figure 1 biomedicines-09-00409-f001:**
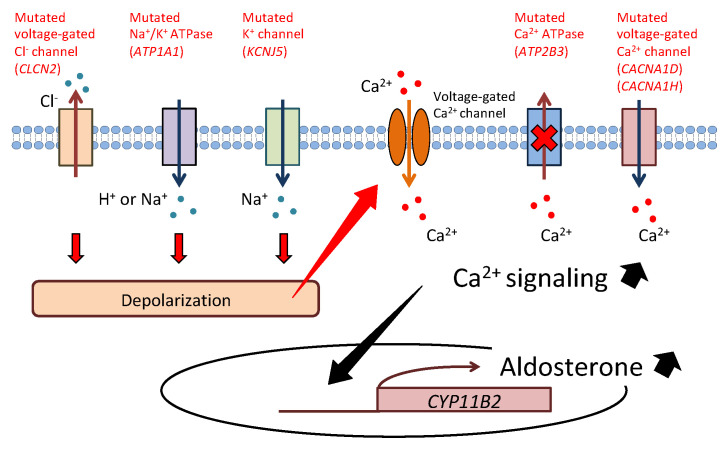
Cellular mechanism of aldosterone synthesis in aldosterone-producing adenoma. Mutations of *KCNJ5*, *ATP1A1,* and *CLCN2* lead to depolarization of the cell membrane due to impairment of ion transport. Depolarization activates voltage-gated Ca^2+^ channels and increases intracellular Ca^2+^ levels. Conversely, mutations of *CACNA1D* and *CACNA1H* directly cause an increase in Ca^2+^ conductance. *ATP2B3* mutation reduces Ca^2+^ export from the cell. Activated calcium signaling promotes transcription of aldosterone synthase (*CYP11B2*), resulting in overproduction of aldosterone.

**Figure 2 biomedicines-09-00409-f002:**
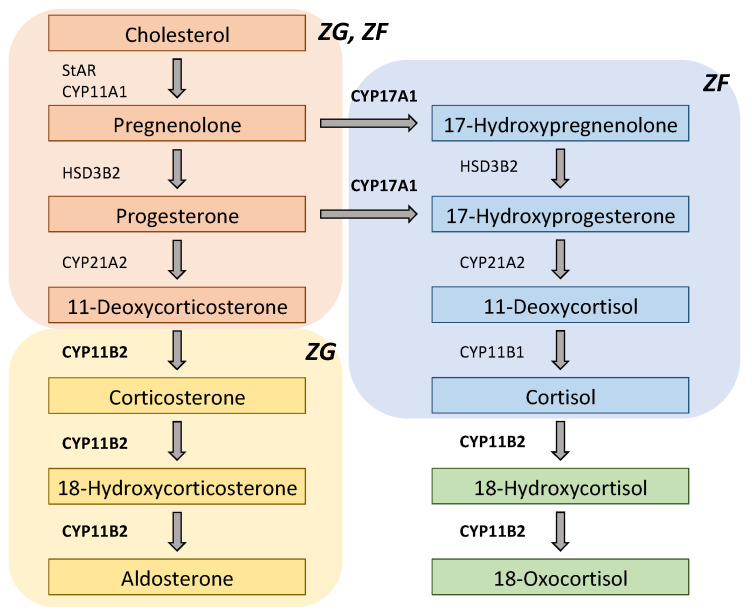
Scheme of steroidogenic pathways for aldosterone, 18-oxocortisol, and 18-hydroxycortisol. Both CYP11B2 (aldosterone synthase) and CYP17A1 (17α-hydroxylase/17,20-lyase) are required to synthesize 18-oxocortisol and 18-hydroxycortisol. Thus, plasma levels of 18-oxocortisol and 18-hydroxycortisol are likely to be higher in patients with *KCNJ5*-mutated aldosterone-producing adenoma (APA), while they are very low in normal adults. CYP11A1: cytochrome P450 cholesterol side-chain cleavage; CYP11B1: 11β-hydroxylase; CYP21A2: 21-hydroxylase; HSD3B2: 3β-hydroxysteroid dehydrogenase type 2; StAR: steroidogenic acute regulatory protein; ZF: zona fasciculata; ZG: zona glomerulosa.

**Table 1 biomedicines-09-00409-t001:** Clinical and histological features of APA harboring each somatic mutation.

Gene	Clinical Characteristics	Histological Features
*KCNJ5*	More common in Asians More often female Diagnosed at younger age Larger tumor size Higher plasma levels of aldosterone, 18-oxocortisol, and 18-hydroxycortisol More likely to have hypertension remission after adrenalectomy	Clear cell dominant (ZF-like)
*ATP1A1*	More often male Smaller tumor size	Compact cell dominant (ZG-like)
*ATP2B3*	More often male Smaller tumor size	Compact cell dominant (ZG-like)
*CACNA1D*	More common in African Americans More often male Smaller tumor size	Compact cell dominant (ZG-like)
*CTNNB1*	More often female Higher risk of post adrenalectomy residual hypertension	Variable

**Table 2 biomedicines-09-00409-t002:** Clinical and molecular characteristics of familial hyperaldosteronism (FH).

	Genetic Variant	Molecular Mechanism	Clinical Characteristics
Type 1	*CYP11B1/CYP11B2* chimeric gene	ACTH induces transcription of *CYP11B2* (coding region)	Glucocorticoid-suppressive hyperaldosteronism
Type 2	*CLCN2* mutations	Increased Cl^-^ efflux activates *CYP11B2* transcription	Early-onset PA
Type 3	*KCNJ5* mutations	Increased Na^+^ influx activates *CYP11B2* transcription	Severe early-onset PA (T158A, I157S, E145Q, G151R) Mild PA (G151E, Y152C)
Type 4	*CACNA1H* mutations	Increased Ca^2+^ influx activates *CYP11B2* transcription	Early-onset PA

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
