# Peer review of "Update on Genetics of Primary Aldosteronism"

_biomedicines, 2021, doi:10.3390/biomedicines9040409_

Round 1

Reviewer 1 Report

The authors tried to overview the pathogenesis of autonomous aldosterone production in PA from the aspects of gene analysis. The present report is well written to understand the mechanisms inducing aldosterone excess in PA patients by analyzing the recent  reports of genetic abnormalities. This is worth informing to the researchers working on pathogenesis of primary aldoteronism, while there are several problems as described below.

Major problems;

  1. The authors need to summarize the relationship between each somatic mutation and pathological characteristics with tumor sizes in Table.
  2. Is it possible to explain when and/or how to acquire each specific somatic mutation?

Also describe the reason of the difference in prevalence of each mutation among races and gender.

  1. I would like to recommend you to summarize each familial hyperaldosteronism aa Table containing clinical characteristics and manifestations for better understanding to the readers.

Reviewer 2 Report

The paper is now suitalbe for publication.

The work accurately describes current knowledge on the subject

Author Response

We sincerely appreciate the detailed reviews of our manuscript.